# Enhanced Efficacy of Radiopharmaceuticals When Using Technetium-99m-Labeled Liposomal Agents: Synthesis and Pharmacokinetic Properties

**DOI:** 10.3390/biomedicines10112994

**Published:** 2022-11-21

**Authors:** Anfal M. Alkandari, Yasser M. Alsayed, Atallah M. El-Hanbaly

**Affiliations:** 1Biophysics and Physics Department, Faculty of Science, Mansoura University, Mansoura 35516, Egypt; 2Nuclear Medicine Department, Faculty of Medicine, KasrAlainy School of Medicine, Cairo University, Cairo 11562, Egypt; 3Theoretical Physics Group, Physics Department, Faculty of Science, Mansoura University, Mansoura 35516, Egypt

**Keywords:** liposome, chromatography, pharmacokinetics, ^99m^Tc, radiopharmaceutical

## Abstract

Challenges posed by the retention of radiopharmaceuticals in unintended organs affect the quality of patient procedures when undergoing diagnostics and therapeutics. The aim of this study was to formulate a suitable tracer encapsulated in liposomes using different techniques and compounds to enhance the stability, uptake, clearance, and cytotoxic effect of the radiopharmaceutical. Cationic liposomes were prepared by a thin-film method using dipalmitoyl phosphatidylcholine (DPPC) and cholesterol. Whole-body gamma camera images were acquired of intravenously injected New Zealand rabbits. Additionally, liposomes were assessed using stability, toxicity, zeta potential, and particle size tests. In the control cases, Technetium-99m (^99m^Tc)-sestamibi exhibited the lowest heart uptake the blood pool and delayed images compared to both ^99m^Tc-liposomal agents. Liver and spleen uptake in the control samples with ^99m^Tc-sestamibi increased in 1-h-delayed images, unlike with ^99m^Tc-liposomal agents, which were decreased in delayed images. The mean maximum count in the bladder for ^99m^Tc-sestamibi loaded liposomes 1 h post-injection was 2354.6 (±2.6%) compared to 178.4 (±0.54%) for ^99m^Tc-sestamibi without liposomes. Liposomal encapsulation reduced the cytotoxic effect of the sestamibi. ^99m^Tc-MIBI-cationic liposomes exhibited excellent early uptake and clearance compared to ^99m^Tc-MIBI without liposomes. Adding cholesterol during liposome formation enhanced the stability and specificity of the targeted organs.

## 1. Introduction

Radionuclide imaging provides an image of the distribution of a radioactively labeled substance within the body after it has been administered to a patient [1]. This is accomplished by recording the radioactive emission using an external radiation detector, a gamma camera, and a PET scanner. The radionuclide is cleared from the body by radioactive decay and biological clearance, the effective half-life [2]. This paper aimed to reduce exposure to radioactivity in the body by enhancing the clearance rate using radioactively labeled liposomal agents.

Liposomes are vesicles composed of two sheets of tightly arranged phospholipids. These molecules have a hydrophobic tail and hydrophilic head regions [3]. The efficacy of liposomes in a drug delivery system can be affected by the number and rigidity of lipid bilayers, as well as their size, surface charge, lipid organization, and surface modification [4].

Generally, liposomes are classified as single or multi lipid bilayers, depending on the number of phospholipid layers [5]. Liposomes can be positively, negatively, or neutrally charged, depending on the heads of the phospholipids. Phospholipids with longer, saturated hydrocarbon chains have an increased ability to interact, thereby forming rigidly ordered bilayer structures, whereas phospholipids with shorter, unsaturated hydrocarbon chains form liposomes with disordered bilayers [6,7]. Phospholipids with longer tails, a low degree of tail unsaturation, ether linkages, cholesterol (CH), propylene glycol, polyethylene glycol (PEG), and polymers such as chitosan, are used to improve the stability of liposomes [8,9,10]. Liposomes are prepared using various methods, such as passive or active loading [10,11].

Lamichhane et al. reviewed a study by Espinola et al., which reported that constant distribution of^111^In-oxine-labeled vesicles was observed over 72 h, compared to ^99m^Tc-labeled vesicles, which showed continuous leakage of radioactivity from the involved organs [12]. A discussion on the possibility of using a radionuclide (ex. ^99m^Tc, ^111^In, iodine, ^18^F, ^52^Mn, ^89^Zr, and ^64^Cu) to radiolabel liposomes was reported by Man et al. Radiolabeling liposomes of different compositions was also considered a convenient way to assess the effect of individual modifications on their whole-body distribution, and such a technique has been in use since the early days of liposomal development [13]. Allard et al. revealed that using lipid nanocapsules (LNC) to entrap lipophilic complexes is a safe and potent antitumor system for the treatment of malignant gliomas, as it prolongs tissue retention and increases the median survival time [14].

However, Datta et al.’s revision suggested that future studies should be conducted on the biocompatibility of radiopharmaceuticals, as neither the material (gold, liposome, polymer, rare-earth, etc.) delivery nor any antibodies used for active targeting should cause an immune or foreign-body reaction, thus confirming that, biodistribution and toxicity tests should be carried out [15]. The authors also recommended that the particle size should ideally be below 400 nm and preferably <40 nm for optimal clearance. Furthermore, they suggested the need for studies to design extracellular vesicles labeled directly with radionuclides to increase therapeutic efficacy [15].

Moreover, a study performed on ^99m^Tc-free liposomes, ^99m^Tc- human serum albumin-liposomes, and ^99m^Tc-nanocolloid-liposomes noted that the heart exhibited satisfactory extraction activity, even with subcutaneous administration; this method was used in the present study to determine the biodistribution of a cardiac tracer combined with lipid nanocapsules injected intravenously and to avoid problems clarified by researchers mentioned above [16].The main advantages of adding liposomes include that they increase the stability of drugs via encapsulation, increase the efficacy and therapeutic index of drugs, reduce the toxicity of the encapsulated agent, and reduce the exposure of sensitive tissues to toxic drugs. Furthermore, they can be used to deliver both hydrophilic and lipophilic drugs, and they can be targeted to facilitate site-specific delivery to tumor tissues.

## 2. Materials and Methods

### 2.1. Materials and Reagents

Technescan^®^Sestamibi (MIBI) tetrakis (2-methoxyisobutylisonitrile) copper (I) tetrafluoroborate (LDRP, Parramatta, Australia), Dipalmitoyl phosphatidylcholine (DPPC) (Merck KGaA, Darmstadt, Germany), cholesterol (Merck KGaA), ammonium sulfate for molecular biology (Merck KGaA), stearyl amine cationic lipid (Merck KGaA), and chloroform (Merck KGaA) were sourced for this research. A Milli-DI^®^water purification system was used to produce deionized water (Merck Millipore, Burlington, MA, USA).

### 2.2. Radiopharmaceutical Samples

Three samples were used in this study: ^99m^Tc-MIBI without liposomes (control sample), ^99m^Tc-MIBI added to lower pH liposomes (^99m^Tc-MIBI-free liposomes gradient technique), and MIBI encapsulated during the formation of the liposome then labeled with ^99m^Tc (^99m^Tc-liposomes loaded with MIBI).

### 2.3. Preparation of Liposomes and Drug Loading

Positively charged liposomes were prepared using 1,2-dipalmitoylphosphocholine (DPPC), cholesterol, and stearyl amine with a molar ratio of 7:2:1. Once the lipids were thoroughly mixed in the organic solvent, the solvent was removed to yield a lipid film. The lipid film was thoroughly dried to remove residual organic solvent by placing the round-bottomed flask on a rotary vacuum evaporator for 30 min at 40 °C and 80 rpm. The dried lipid film was then hydrated by ammonium sulfate at pH 5.0, yielding a 1mg/mL sample of DPPC. Then, the hydrated lipid suspension was downsized by sonication in a bath sonicator and homogenized for 10 min at 20,000 rpm using a shear homogenizer. This method is called thin-film hydration, which is considered the simplest and most practicable technique to produce liposomes. Finally, MIBI was added at a ratio of 1:1 (W/W) to the DPPC during the formation of the liposomes [17,18,19]. The samples were then analyzed using high-performance liquid chromatography (HPLC) to ensure encapsulation efficiency [20].

### 2.4. Analytic Methods

The HPLC system (Nouryon, Bohus, Sweden) was equipped with a C18-bonded silica column and silica-based monomeric-type reversed-phase (RP) packing material (Kromasil, C18, 100 Å pore size, 150 × 4.6 mm I.D., 5 µm particle size). The sample was dissolved in a portion of the mobile phase, at 98:2%, including 0.1% trifluoroacetic acid (TFA) in water/acetonitrile, methyl cyanide (MeCN), using five different concentrations. The flow rate was 1 mL/min and we used a 260 nm UV detector. The particle size and zeta potential of the nanoparticles were also tested by dynamic light scattering (DLS) (Zetasizer Nano ZN, Malvern Panalytical Ltd., Malvern, UK) [10,20].

### 2.5. Cytotoxic Activities

An (H9C2) rat heart/myocardium was obtained (American Type Culture Collection, ATCC, Manassas, VA, USA). Cells were maintained in DMEM (Dulbecco’s modified eagle medium) supplemented with 100 mg/mL of streptomycin, 100 units/mL of penicillin and 10% of heat-inactivated fetal bovine serum in a humidified 5% (*v/v*) CO_2_ atmosphere at 37 °C. Cell viability was assessed by a sulforhodamine B SRB assay. Aliquots of 100 μL cell suspension (5 × 10^3^ cells) were seeded in 96-well plates and incubated in complete media for 24 h. Cells were treated with another aliquot of 100 μL media containing drugs at various concentrations. After 72 h of drug exposure, cells were fixed by replacing media with 150 μL of 10% trichloroacetic acid (TCA) and incubated at 4 °C for 1 h. The TCA solution was removed, and the cells were washed 5 times with distilled water. Aliquots of 70 μL SRB solution (0.4% *w/v*) were added and incubated in a dark place at room temperature for 10 min. Plates were washed 3 times with 1% acetic acid and allowed to air-dry overnight. Then, 150 μL of TRIS (10 mM) was added to dissolve protein-bound SRB stain; the absorbance was measured at 540 nm using a BMG LABTECH^®^- FLUOstar Omega microplate reader (Ortenberg, Germany).

### 2.6. Labeling Procedure

Technetium ^99m^sestamibi (MIBI), known as ^99m^Tc-methoxy isobutyl isonitrile, is a lipophilic cationic radiotracer used as a radioactive diagnostic agent to image cardiac, breast, and parathyroid tissues. All rabbits were injected intravenously with 74 MBq/kg of^99m^Tc-MIBI. The first sample (control) contained no liposomes, whereas a ^99m^Tc-MIBI kit was prepared in a boiling water bath. In the second sample, ^99m^Tc-MIBI was encapsulated within free cationic liposomes at pH 5.0 via the pH gradient technique. The other sample was MIBI encapsulated within liposomes during the formation of cationic liposomes’, then added to a ^99m^Tc elution [21].

The radiochemical purity (RCP) of the technetium ^99m^Tc sestamibi was determined by thin-layer chromatography (TLC) to separate impurities, mainly free pertechnetate (^99m^$Tc_{4}^{-}$), to ensure that the impurities did not interfere with the quality of the image or result in an unacceptably high radiation dose to the patient (e.g., stomach and thyroid gland). The radiochemical yield (96.4%) of ^99m^Tc-MIBI was obtained at room temperature at pH = 7.4. Radiopharmaceutical TLC was performed using a sheet of plastic coated with a thin adsorbent material layer, usually silica gel or aluminum oxide (stationary phase) (Merck Millipore). A drop of the radiopharmaceutical was placed 2 cm from the bottom of the strip. The strip was placed in a tube containing the mobile phase (solvent: ethanol); however, the spot must remain above the solvent level. The solvent was allowed to migrate up the strip by capillary force until it reached the pre-marked solvent front; then, the strip was removed and analyzed using a NaI(Tl) scintillation counter (Atomlab™ 500 Dose Calibrator Biodex, New York, NY, USA).

### 2.7. Dosimetry and Imaging Protocol

In this study 74, MBq/kg of radiopharmaceutical was injected intravenously through amarginal ear vein. A total of 296 MBq^99m^Tc-MIBI and ^99m^Tc-liposomal agents diluted in 5 mL saline was administered. Images were acquired with a double-headed gamma camera (Symbia, Siemens health care, IL, USA) and a low-energy, high-resolution parallel whole collimator in whole-body (WB) scanning mode with a 128 × 128 matrix at a zoom factor of 1.45 and 300 s per WB image after 5 min, 1 h, and 24 h. The energy windows were set to 140 keV ± 20%, and the images were obtained immediately and 1 h and 24 h after tracer injection. The region of interest (ROI) was manually drawn on WB images in all the acquisitions to determine the organ activity at each time point.

### 2.8. Animal Model

All animal studies were performed according to protocols approved by the institutional animal care committee and followed the guidelines for animal care on housing, husbandry, and pain management services. Five adult male New Zealand rabbits with a mean weight of approximately 4 kg were sampled in the study. The rabbits were anesthetized intramuscularly with a ketamine: xylazine mixture (105 and 15 mg, respectively). The same five rabbits were studied for the three samples (^99m^Tc-MIBI control sample, ^99m^Tc-MIBI-free liposome gradient technique, and ^99m^Tc-liposomes loaded with MIBI); each was sampled a week apart to ensure the elimination of radioactivity before the next radiopharmaceutical administration.

### 2.9. Data Analysis

Regions of interest (ROIs) were drawn to evaluate the biodistribution, activity uptake, and clearance rate of the radiopharmaceuticals. The standard error of the mean (SEM), along with the mean, was used to report the statistical analysis results.

## 3. Results

After many trials, the formula with the smallest particle size was selected. Trisodium citrate and ammonium sulfate were tested individually as hydration buffers with concentrations of 10, 100, and 300 mM and pH between 3.0 and 5.5. The smallest particle size resulted from 10 mM of ammonium sulfate with pH 5.0. The MIBI_ liposomes were prepared using fewer cationic particles, as the MIBI was already positively charged. The use of sonication followed by homogenization produced smaller particles sizes than homogenization followed by sonication. The resulting liposome size was 174.4 ± 2.954 nm with a zeta potential of 33.6 ± 3.23 mV.

A standard calibration curve was constructed with five different concentrations of MIBI encapsulated in positive liposomes before ^99m^Tc labeling to ensure the encapsulation efficiency and the accuracy of the method: 0.00076, 0.076, 0.76, 7.6, and 760 ng/mL were obtained with a flow rate of 1 mL/min using HPLC (Figure 1). The linear relationship of the concentrations and the high value of R^2^ = 0.99957 were derived from equation y = 80,892.61502x + 3725.85534, with an entrapment efficiency percentage of 99.9997%.

A toxicity test was also performed, and the half-maximal inhibitory concentration (IC50) was found to be >100 µg/mL. The dose response and cell viability of neutral and positively charged liposomes, MIBI encapsulated within liposomes, and the tracer MIBI are shown in the images in Figure 2. The MIBI toxicity test revealed that IC50 was at a concentration of 122 µg/mL, with a cell survival rate of 59%. However, the proportion of viable cells exposed to encapsulated MIBI within liposomes reached 70.5%. A total of five rabbits were included in the study. All were scanned immediately after radiopharmaceutical intravenous injection, as well as 1 h and 24 h post-injection. All ^99m^Tc liposomal agents were assessed by radiochemical purity (RCP) > 95% using thin-layer chromatography (TLC) to test the labeling efficiency and ensure the absence of impurities.

Figure 3 illustrates the biodistribution and clearance in the (a) heart, (b) bowel, (c) liver, and (d) spleen of the ^99m^Tc-MIBI-free liposomes and ^99m^Tc-MIBI-loaded cationic liposomes compared to ^99m^Tc-MIBI as a control sample. Table 1 shows the uptake of ^99m^Tc-MIBI, ^99m^Tc-free liposome-MIBI, and ^99m^Tc-encapsulated MIBI in the heart, liver, spleen, bowel, kidneys, and bladder in blood-pool images. The heart blood-pool uptake in the ^99m^Tc-liposomes loaded with MIBI and ^99m^Tc-MIBI-free liposomes was greater than standard ^99m^Tc-MIBI by 800% and 560%, respectively. Furthermore, the heart activity in blood-pool images was higher for liposomes loaded with MIBI than for MIBI-free liposomes. The activity clearance of ^99m^Tc-MIBI, ^99m^Tc-free liposomes-MIBI, and ^99m^Tc-encapsulated MIBI in the heart, liver, spleen, bowel, kidneys, and bladder 1 h post-injection is described in Table 2. A significant relationship with heart uptake was observed in ^99m^Tc- MIBI-free liposomes in the blood-pool and 1-h-delayed images (*p* < 0.05). Table 3 shows 24 h post injection pharmacokinetics of ^99m^Tc-MIBI, ^99m^Tc-free liposome-MIBI, and ^99m^Tc-encapsulated MIBI in the heart, liver, spleen, bowel, kidneys, and bladder. The heart activity for both ^99m^Tc liposomal agents started to decrease after the blood-pool images; however, for ^99m^Tc-MIBI, the heart activity continued to increase after the blood-pool, with the decrease starting 1 h post-injection. We also observed a significant relationship with respect to heart uptake between ^99m^Tc-MIBI and ^99m^Tc-MIBI-free liposomes 24 h post injection.

In a univariate logistical analysis, the value (*p* < 0.05) was significant when comparing the bowel activity of 24 h delayed and 1 h delayed images and using ^99m^Tc-liposomes loaded with MIBI [22]. The bowel activity in 24-h-delayed images with ^99m^Tc-liposomes loaded with MIBI was significantly less than for ^99m^Tc-MIBI-free liposomes (*p* ≤ 0.05). The bowel activity in the control samples (^99m^Tc-MIBI) for the blood-pool images immediately increased 1 h and 24 h post injection, increasing over time (1.14% ± 0.08% SEM, 2.79% ± 0.33% SEM, and 2.93% ± 0.15% SEM, respectively). However, the bowel activity started to decrease for both the liposomes loaded with MIBI and the MIBI-free liposomes 1 h post-injection. There was a significant difference in the ^99m^Tc-MIBI-free liposome bowel uptake between 1 h and 24 h post-injection. Nonetheless, there was a significant difference in the blood-pool bowel uptake levels between the ^99m^Tc-MIBI control and the ^99m^Tc-MIBI-free liposomes (*p* < 0.05).

The liver uptake was the highest in the blood pool images for both ^99m^Tc-liposomal agents, whereas the highest uptake occurred 1 h post-injection for the ^99m^Tc-MIBI, then decreased. At 1 h post-injection, the liver uptake percentage for ^99m^Tc-MIBI was 2.9% ± 0.4; for ^99m^Tc-MIBI-free liposomes, it was 2.68% ± 0.33; and for ^99m^Tc-liposomes loaded with MIBI, it was 2.02% ± 0.19. Furthermore, this study also showed a significant difference (*p* < 0.05) in the blood-pool liver uptake between ^99m^Tc-MIBI-free liposomes and ^99m^Tc-liposomes loaded with MIBI. Also, there was a statistically significant relationship between the ^99m^Tc-MIBI without liposomes and the ^99m^Tc-MIBI-free liposomes 1 h post injection in terms of their uptake in the liver 24 h post injection. Moreover, for the liver uptake at 24 h post-injection, there was a significant association between the ^99m^Tc-MIBI, control, and ^99m^Tc-MIBI-free liposomes. Regarding the bias-to-variance-characteristics (BVCs), the spleen uptake demonstrated no significant difference in the blood-pools for the ^99m^Tc-liposomes loaded with MIBI and the control ^99m^Tc-MIBI samples (*p* = 0.6).

As shown in Figure 4, anterior whole-body images of the rabbits were acquired using a dual-head gamma camera. For all liposomal agents, blood pool images showed high radiopharmaceutical extraction, and delayed images showed fast washout. Figure 5 compares the clearance in the kidneys and bladder for the control ^99m^Tc-MIBI and the ^99m^Tc-liposomal agents. The mean of the maximum counts for kidney activity in the blood-pool images for ^99m^Tc-MIBI without liposomes was lower than that for both ^99m^Tc-liposomal agents. At 1 h post injection, the mean maximum count in the kidneys for both liposomal agents was close to the average in the control samples without liposomes. However, the mean maximum count in the bladder for ^99m^Tc-liposomes loaded with MIBI and ^99m^Tc-MIBI-free liposomes 1 h post injection was higher than for the ^99m^Tc-MIBI control sample. Early kidney images of ^99m^Tc-MIBI-free liposomes showed significantly greater uptake than ^99m^Tc-MIBI (*p* ≤ 0.001).

## 4. Discussion

In this study, we found that both ^99m^Tc-liposomal agents, with MIBI encapsulated during liposome formation and by the pH gradient technique, enhanced the biodistribution, uptake, and clearance of radiopharmaceuticals, which may reduce their toxicity. A cytotoxic test of MIBI confirmed that IC50 was at a concentration of 122 µg/mL with a cell survival rate of 59%, whereas the proportion of viable celled encapsulated with-MIBI within liposomes was 70.5% at the same concentration. This finding supports the idea that liposomes are not only safe but also decrease the toxic effect of the encapsulated tracer, thereby increase the cell survival rate.

In general, MIBI tended to be trapped in the heart and liver, with less distribution in the spleen and bowel. The uptake of ^99m^Tc-liposomal agents, ^99m^Tc-MIBI-free liposomes and ^99m^Tc-liposomes loaded with MIBI in blood-pool images was higher than for the control ^99m^Tc-MIBI. A comparison of the liposomal agents to the control radiopharmaceutical agent shown that the uptake was faster, and the extraction was higher immediately after dose administration. The results also demonstrate that the ^99m^Tc-MIBI blood-pool heart uptake was influenced by the liposome encapsulation, as the maximum count was higher for liposomes loaded with MIBI than for MIBI-free liposomes. Meanwhile, there was little change in the bowel uptake for ^99m^Tc-liposomes loaded with MIBI in the blood pool and 1 h post injection, which means that the heart had sufficient uptake and was not obscured by the presence of artifacts. The liver and spleen uptake in control samples (^99m^Tc-MIBI) increased in 1 h-delayed images, unlike ^99m^Tc-MIBI-free liposomes and ^99m^Tc-liposomes loaded with MIBI.

A further novel finding is that 1 h post injection bladder activity increased with the ^99m^Tc-liposomes loaded with MIBI, but the same was not true for kidney activity. In comparison, ^99m^Tc-MIBI without liposomes had resulted in almost the same level of kidney uptake, with much lower bladder uptake. This indicates that after 1 h, the ^99m^Tc-MIBI had just started to clear, whereas the ^99m^Tc-liposomes loaded with MIBI showed fast clearance, resulting from the encapsulation of positive liposomes. Such a fast washout of radioactivity is beneficial as it lowers the radiation exposure for patients, the public, and healthcare workers if they spend extended periods with injected patients taking extra images or delayed scans.

For imaging applications, pharmacokinetically, the blood residency time and tissue uptake of liposomes depend on the size, surface chemistry and charge of the nanoparticles. Parenteral administration of agents below the glomerular filtration cutoff (30–50 kDa or a diameter of approximately 5 nm) means that they are rapidly excreted by the kidneys. Larger nanoparticles may circulate for longer, but they regularly accumulate in the liver and spleen, whereas micron-sized particles become trapped in capillary beds and the pulmonary vasculature [23]. Thus, when comparing the clearance results from this study with results for the same liposomes but of a larger size, i.e., more than 200 nm, the clearance rate became higher, and the liver and spleen uptake were lower [24]. This is also consistent with the results of previous studies involving the radiolabeling of nanoparticles [25,26,27].

On the other hand, the surfaces of the NPs may be modified by the polysaccharide or polymer covering, which may decrease their recognition by passive and active clearance mechanisms [23]. Using stealth liposomes in this way increases the stability and prolongs the circulating half-life, meaning there will be greater uptake at sites of interest. Given the undesirable characteristics of high-energy isotopes, direct labeling of stealth liposomes would be useful for improved and prolonged tracking.

As previously mentioned, Espinola et al. reported continuous leakage of radioactivity-labeled vesicles from the involved organs [12]. In this study the leakage problem was successfully overcome by using a pH gradient technique, as revealed by the HPLC encapsulation efficiency results using five different concentrations. The method was characterized bygood linearity, sensitivity, and specificity. Additionally, the size of the liposomes was reduced for improved clearance. It is also beneficial to use small liposomes for cancer treatment, inflammation therapy, and imaging by exploiting the enhanced permeability retention (EPR) effect [28]. Small liposomes are useful, as they concentrate on the targeted tissues and not the surrounding healthy tissues [28].

Encapsulating MIBI within liposomes improves their uptake and clearance; imaging preparation must be performed to avoid leakage from the liposomes and to ensure their stability. An advantage of loading the MIBI using the pH gradient technique in positive liposomes is that it is possible to use previously prepared lyophilized liposomes with a low pH. Loading radioactive material with a long half-life during liposome preparation is also possible. Given the risk of device contamination, ^99m^Tc was not loaded within liposomes because of its short half-life of 6 h. With encapsulation, the lipid mixture must be heated above the temperature of phase transition for maximum labeling, so for tracers that should not be exposed to heat the pH gradient technique presents a solution.

## 5. Conclusions

In this study, we described the feasibility of using ^99m^Tc-MIBI encapsulated within liposomes as a radiopharmaceutical imaging agent. Two methods were used to load ^99m^Tc-MIBI within liposomes, and both showed excellent stability, pharmacokinetics, and cytotoxicity enchantment compared to ^99m^Tc-MIBI. Liposomes encapsulation with the pH gradient technique is more applicable than tracer loading using the liposome formation method. Based on the findings presented here, future research should investigate how best to use cationic liposomes in nuclear medicine for both the diagnostic and therapeutic fields.

## Figures and Tables

**Figure 1 biomedicines-10-02994-f001:**
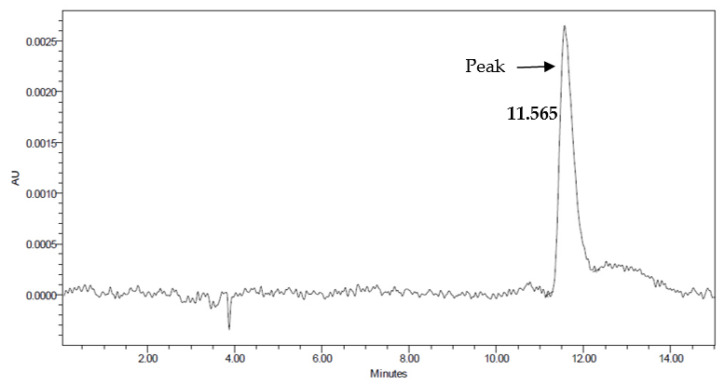
Chromatogram using an HPLC system with Kromasil^®^ packing material (C18, 100 Å pore size, 150 × 4.6 mm I.D., 5 µm particle size) for analysis of encapsulation efficiency of MIBI encapsulated in positive liposomes before ^99m^Tc labeling at a concentration of 0.76 ng/mL. Mobile phase, at 98:2%, including 0.1% trifluoroacetic acid in water/acetonitrile. The flow rate was 1 mL/min. A 260 nm UV detector was used. The *x*-axis represents the retention time, and the *y*-axis represents the absorbance units (AUs).

**Figure 2 biomedicines-10-02994-f002:**
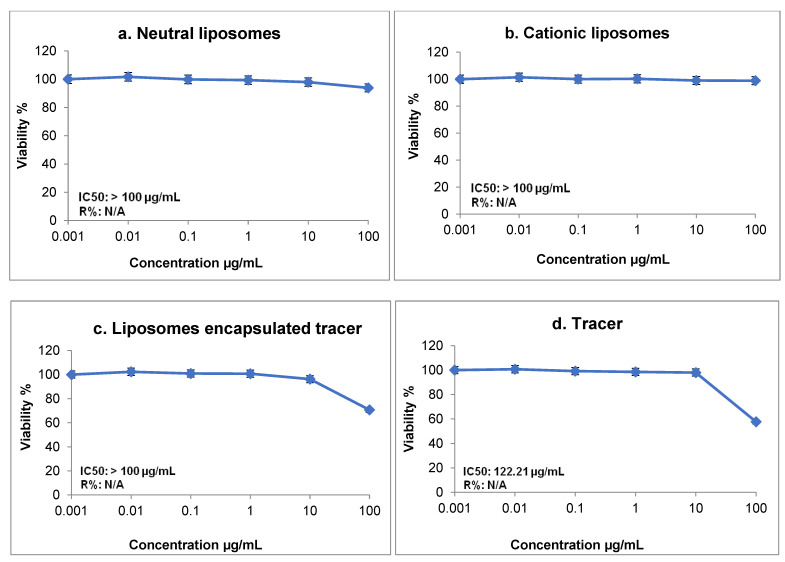
Dose-response curves assessed by SRB showing the cytotoxic activities of (**a**) neutral empty liposomes (**b**) cationic empty liposomes, and (**c**) liposomes encapsulated with tracer MIBI, as well as the effect of (**d**) control, tracer MIBI, on the cell viability of an (H9C2) rat heart/myocardium. Cells were maintained in DMEM supplemented with 100 mg/mL of streptomycin, 100 units/mL of penicillin, and 10% heat-inactivated fetal bovine serum in a humidified 5% (*v/v*) CO_2_ atmosphere at 37 °C. Aliquots of 100 μL cell suspension (5 × 10^3^ cells) were seeded in 96-well plates and incubated in complete media for 72 h.

**Figure 3 biomedicines-10-02994-f003:**
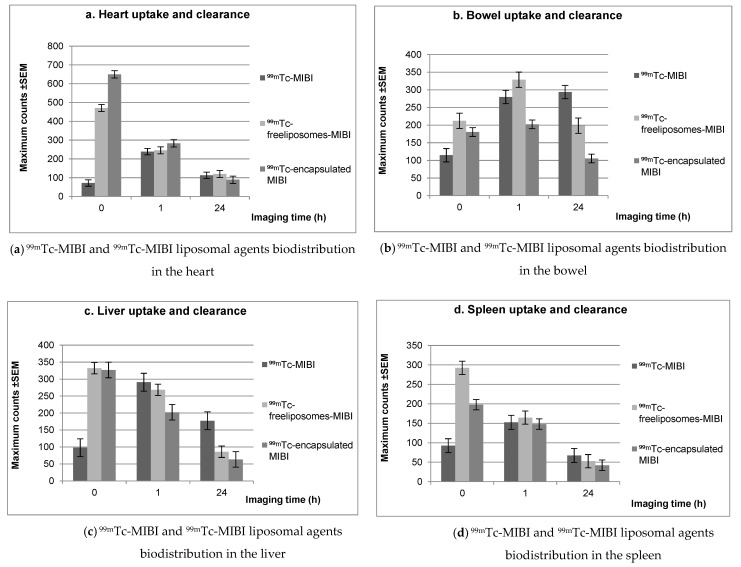
^99m^Tc-MIBI, ^99m^Tc-free liposome-MIBI, and ^99m^Tc-encapsulated MIBI pharmacokinetics in the heart, bowel, liver, and spleen in the blood-pool, 1 h, and 24 h post-injection images using ROI maximum counts ± standard error of mean.

**Figure 4 biomedicines-10-02994-f004:**
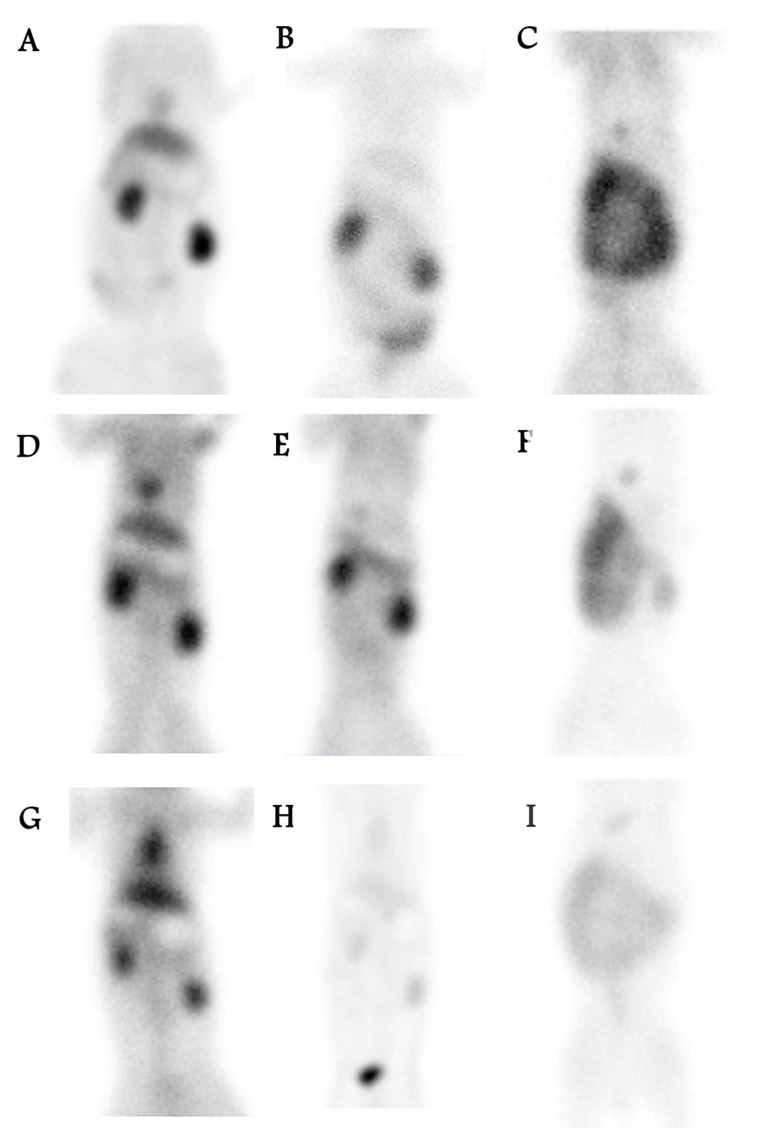
Anterior whole-body images of rabbits acquired using a dual-head gamma camera. (**A**) ^99m^Tc-MIBI blood-pool. (**B**) ^99m^Tc-MIBI delayed 1 h (**C**) ^99m^Tc-MIBI delayed 24 h (**D**) ^99m^Tc-MIBI-free positive liposomes blood-pool. (**E**) ^99m^Tc-MIBI-free positive liposomes delayed 1 h. (**F**) ^99m^Tc-MIBI-free positive liposome delayed 24 h. (**G**) ^99m^Tc-loaded with MIBI within positive liposomes blood pool. (**H**) ^99m^Tc-loaded with MIBI within positive liposomes delayed 1 h. (**I**) ^99m^Tc loaded with MIBI within the positive liposomes delayed 24 h.

**Figure 5 biomedicines-10-02994-f005:**
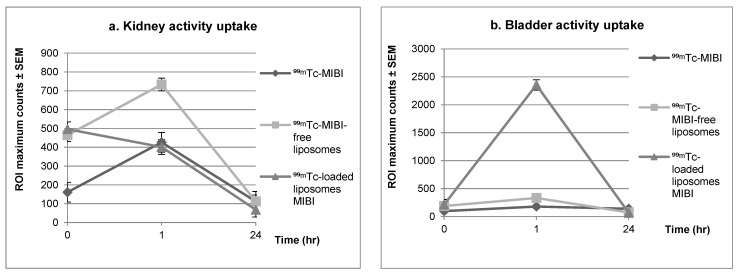
ROI maximum counts ± standard error of mean of (**a**) kidneys and (**b**) bladder for comparison of the clearance of the control ^99m^Tc-MIBI, ^99m^Tc-MIBI-free liposomes, and ^99m^Tc-MIBI-loaded liposomes in blood-pool images immediately after dose administration, as well as 1 h and 24 h post injection.

**Table 1 biomedicines-10-02994-t001:** Blood-pool biodistribution of ^99m^Tc-MIBI compared to that of ^99m^Tc-liposomal agents in different organs.

Organ	^99m^Tc-MIBI	^99m^Tc-Free Liposome- MIBI	^99m^Tc-Liposome Encapsulated MIBI
Heart	71.4 ± 0.12	471 ± 0.3	649.4 ± 0.27
Liver	97.8 ± 0.26	332 ± 0.11	326.8 ± 0.4
Spleen	92.4 ± 0.2	292.4 ± 0.25	197.8 ± 0.17
Bowel	114.6 ±0.08	212.2 ± 0.08	180.4 ± 0.087
Kidneys	160.9 ± 0.26	464.5 ± 0.41	496.1 ± 0.48
Bladder	96.8 ± 0.18	187.6 ± 0.14	211.4 ± 0.06

^99m^Tc-MIBI, ^99m^Tc-free liposome-MIBI, and ^99m^Tc-encapsulated MIBI biodistribution in the heart, liver, spleen, bowel, kidneys, and bladder in blood-pool images using ROI maximum counts ± standard error of mean percent.

**Table 2 biomedicines-10-02994-t002:** Biodistribution of ^99m^Tc-MIBI compared to that of ^99m^Tc-liposomal agents in different organs at 1 h post-injection.

Organs	^99m^Tc-MIBI	^99m^Tc-Free Liposome-MIBI	^99m^Tc-Liposome Encapsulated MIBI
Heart	238.4 ± 0.25	245.4 ± 0.21	282.8 ± 0.25
Liver	290.8 ± 0.24	268.2 ± 0.33	202.2 ± 0.19
Spleen	152.4 ± 0.15	164.6 ± 0.16	148 ± 0.16
Bowel	279.6 ± 0.33	328.6 ± 0.18	202 ± 0.17
Kidneys	426.3 ± 0.12	733.6 ± 0.45	401 ± 0.52
Bladder	178.4 ± 0.54	332.2 ± 0.24	2354.6 ± 2.6

^99m^Tc-MIBI, ^99m^Tc-free liposome-MIBI, and ^99m^Tc-encapsulated MIBI biodistribution in the heart, liver, spleen, bowel, kidneys, and bladder in 1 h post-injection images using ROI maximum counts ± standard error of mean percent.

**Table 3 biomedicines-10-02994-t003:** Biodistribution of ^99m^Tc-MIBI compared to ^99m^Tc-liposomal agents in different organs 24 h post injection.

Organs	^99m^Tc-MIBI	^99m^Tc-Free Liposome-MIBI	^99m^Tc-Liposome Encapsulated MIBI
Heart	112.4 ± 0.16	119.6 ± 0.03	89 ± 0.19
Liver	177.2 ± 0.28	86 ± 0.05	63.4 ± 0.23
Spleen	67.2 ± 0.12	52.6 ± 0.08	42.2 ± 0.06
Bowel	293.8 ± 0.15	198.4 ± 0.39	105.4 ± 0.1
Kidneys	112.1 ± 0.05	112.5 ± 0.15	68.1 ± 0.15
Bladder	139.2 ± 0.25	73.4 ± 0.06	67.8 ± 0.09

^99m^Tc-MIBI, ^99m^Tc-free liposome-MIBI, and ^99m^Tc-encapsulated MIBI biodistribution in the heart, liver, spleen, bowel, kidneys, and bladder in 24 h post-injection images using ROI maximum counts ± standard error of mean percent.

## Data Availability

All data generated or analyzed during this study are included in this published article.

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
