# Peer review of "Enhanced Efficacy of Radiopharmaceuticals When Using Technetium-99m-Labeled Liposomal Agents: Synthesis and Pharmacokinetic Properties"

_biomedicines, 2022, doi:10.3390/biomedicines10112994_

Round 1

Reviewer 1 Report

Interesting research showing a different in vivo behaviour of 99m Tc-MIBI and liposomes containing the radiopharmaceutical Tc-labelled after preparation or encapsulated.

The most interesting result is the Heart/Liver and Heart/Bowel ratios that at 1 hr p.i (that is the time of a diagnostic scan) are 1.41 and 1.39 for MIBI-encapsulated liposomes, higher in comparison to the other formulations.

            However some points need to be elucidated:

* In the text of pag 8 the aa report data expressed as % but it is unclear its meaning.  % / injected dose? but in this case it is impossible to calculate from planar views. Please explain the method employed.

* In the section Dosimetry and imaging protocol it seems that total cts in the organ ROI are reported but in fig.3  results are expressed as max cts.  Probably these results are decay corrected, but it is not specified

* It is not  reported how the free pertechnetate is removed from the medium after labelling (at room temperature?) and which is the labelling yield

            Minor comments:

*Title: it is preferable to insert the word   “sestamibi” in the title

*Fig.4: grey scale is not explained: max activity at the top? same setting for all images (corrected per decay)?

*Fig 5: different colors in a and b for MIBI-loaded and MIBI free liposomes, may be confusing

*References: Citation style is not uniform

*Abbreviations: preferable in alphabetic order

*Mistyping errors: several in the text (capital, space, bold, underline etc)

Author Response

Response to Reviewers’ Comments

Journal: Biomedicines

Manuscript number: biomedicines-2016488

Manuscript title: Enhanced Efficacy of Radiopharmaceuticals When Using Technetium-99m-Labeled Liposomal Agents: Synthesis and Pharmacokinetic Properties.

Dear Respected Editors/ Reviewers

The authors are highly appreciated the comprehensive feedback through the review process.  Those comments are all valuable and very helpful for revising and improving our paper and researches. Revised portion are marked in yellow in the manuscript. kindly see the attachment to revise the edited manuscript. The main corrections in the paper and the responds to the reviewer comments are as following:

Comments and Suggestions for Authors 

Interesting research showing a different in vivo behaviour of 99m Tc-MIBI and liposomes containing the radiopharmaceutical Tc-labelled after preparation or encapsulated.

The most interesting result is the Heart/Liver and Heart/Bowel ratios that at 1 hr p.i (that is the time of a diagnostic scan) are 1.41 and 1.39 for MIBI-encapsulated liposomes, higher in comparison to the other formulations.

Authors’ response: We thank the reviewer for such thoughtful review of our work and kind words.

However some points need to be elucidated:

* In the text of pag 8 the aa report data expressed as % but it is unclear its meaning.  % / injected dose? but in this case it is impossible to calculate from planar views. Please explain the method employed.

Authors’ response: We agree. In this section we used ROI maximum count percent. For example the maximum count of bowel in blood-pool image using 99mTc-MIBI (control sample) was 114.6 ±0.08 standard error of mean percent. Thus the percentage of the maximum count is 1.14% ± 0.08% SEM.

* In the section Dosimetry and imaging protocol it seems that total cts in the organ ROI are reported but in fig.3  results are expressed as max cts.  Probably these results are decay corrected, but it is not specified.

Authors’ response: In the Dosimetry and imaging protocol section, we mentioned the total activity of the injected dose. But in fig. 3 results we were comparing the biodistribution of each organ by using ROI maximum counts ± standard error of mean.

* It is not  reported how the free pertechnetate is removed from the medium after labelling (at room temperature?) and which is the labelling yield

Authors’ response: We thank the reviewer for this observation. Radiochemical yield is added to the article, kindly see line 146. We ensured the absence of the free pertechnetate from the radiochemical purity test (RCP) > 95% using thin-layer chromatography (TLC). Also in imaging the thyroid gland and stomach were not detected which confirm free pertechnetate absence in the radiopharmaceutical agents.

Minor comments:

*Title: it is preferable to insert the word   “sestamibi” in the title

Authors’ response: We highly respect the reviewer’s concern and we agree with it; however it might not be possible to change the submitted title.

*Fig.4: grey scale is not explained: max activity at the top? same setting for all images (corrected per decay)?

Authors’ response: The first line (horizontally) A, B, and C images represent 99mTc-MIBI (control sample) at blood-pool, 1 hour, and 24 hours post injection. The second horizontal row D, E, and F represent images of 99mTc-MIBI-free positive liposomes. The third horizontal line G, H, and I represent images of 99mTc-loaded MIBI within positive liposomes. Also images are decay-corrected, using the same acquisition settings, and for reporting processes we used syngo.via imaging software with Siemens imaging hardware. The gamma camera origin is added to the article in the Dosimetry and imaging protocol section (see line 158). We hope this answers the reviewer concern.

*Fig 5: different colors in a and b for MIBI-loaded and MIBI free liposomes, may be confusing

Authors’ response: We thank the reviewer for the observation and correction. Fig. 5 colors have been changed as requested, kindly see line 344.

*References: Citation style is not uniform

Authors’ response: We thank the reviewer for pointing this out. The manuscript was revised and adjusted according to this comment. Citation style is edited to ACS style as recommended in the journal’s guideline, and the manuscript has also been comprehensively edited by MDPI English pre-edited services to avoid any error.

*Abbreviations: preferable in alphabetic order

Authors’ response: Thank you for pointing this out. Abbreviation are corrected and listed in alphabetic order.

*Mistyping errors: several in the text (capital, space, bold, underline etc)

Authors’ response: We agree and we thank the reviewer for this observation. The manuscript it edited and corrected accordingly.

Reviewer 2 Report

The submitted manuscript deals with a very interesting topic from the point of view of medical research, which aims to make the use of already established radiopharmaceuticals more efficient. Using the example of [99mTc]sestamibi, the use of liposomes and the influence of uptake kinetics in the monitored tissue is demonstrated.

However, the manuscript is written hastily and contains a number of nomenclatural and factual errors and inaccuracies, as well as slang expressions.

What do the authors mean, the claim that radioactivity is cleared (released)... p.1, paragraph 1? Radioactivity is a process, a substance or a radionuclide can be released. In this sense, the term is completely misleading and should be set in perspective.

What is the chemical form of radionuclides that can leave liposomes? Are they really free radionuclides, as stated on p. 2, paragraph 2? Or can labeled molecules also leave liposomes? Or will radiolysis occur due to nuclear reflection? Could you comment this?

Please pay attention to the correct chemical nomenclature, according to IUPAC, especially writing the nomenclature of labeled molecules. Also, pay attention to the correct writing of indices for radionuclides - in many places this requires checking. The text thus has a very distracting effect and gives the impression of a hastily written manuscript.

e.g. 1,2-dipalmitoyl-sn-glycero-3-phosphocholine ... what does mean -sn-?

also should be 99mTc instead of 99mTc and in case of selectively labeled compounds ought to be [99mTc]. Also ionic form must be written in proper form.

Basic information about the activity of the prepared [99mTc]MIBI complex is missing in the paragraph on labeling. The radiochemical yield is not indicated either. It is completely unacceptable to state the unit of specific activity Mbq/Kg, instead of the correct form MBq/kg.

In paragraphs concerning the animal model: Could you specify a number of subject in animal cohort? Is it correct to perform data only on 5 animals? This is also necessary to comment it due to statistical evaluation of data.

Could you please comment a physical meaning of the entrapment efficiency 99.9997% How did you carry on? On Figure 1 is chromatogram of liposomes without any hot label. How did you carry on labeling or liposomal internalisation of [99mTc]MIBI? How you can be sure, that [99mTc]MIBI remains in liposome for certain time? Evaluation of stability in vitro is missing. In description of TLC the mobil phase and stationary phase are not listed as well as set up of detector and used collimator.

The values in tables and graphs (Table 1, 2, 3 and Figure 3) are in counts. Should be more appropriate to show % of applied dose per mass of organ which is more comon for ex vivo. However, for SPECT would be interesting SUV and related parameters.

The size of used liposome is being described. Did you observe any changes of hydrodynamic ratio of Zeta potencial after internalisation of [99mTc]MIBI?

The conclusion is insuficient. Unfortunately, the conclusion misses the key facts that were reached in the study. I recommend rewriting it.

For the above listed reasons, I recommend accepting the manuscript only after major revisions have been made

Author Response

Response to Reviewers’ Comments

Journal: Biomedicines

Manuscript number: biomedicines-2016488

Manuscript title: Enhanced Efficacy of Radiopharmaceuticals When Using Technetium-99m-Labeled Liposomal Agents: Synthesis and Pharmacokinetic Properties.

Dear Respected Editors/ Reviewers

Thank you very much for taking the time to review our manuscript and for the constructive comments. Authors are grateful for insightful comments and valuable improvement to our paper. We have made the requested changes to the manuscript and hope that these changes have addressed your concerns and comments. Kindly see the attachment for the edited manuscript. 

Comments and Suggestions for Authors

The submitted manuscript deals with a very interesting topic from the point of view of medical research, which aims to make the use of already established radiopharmaceuticals more efficient. Using the example of [99mTc]sestamibi, the use of liposomes and the influence of uptake kinetics in the monitored tissue is demonstrated.

However, the manuscript is written hastily and contains a number of nomenclatural and factual errors and inaccuracies, as well as slang expressions.

Authors’ response: We thank the reviewer for his thoughtful and thorough comments to make our paper more balanced, and we hope that this editing is satisfactory.

What do the authors mean, the claim that radioactivity is cleared (released)... p.1, paragraph 1? Radioactivity is a process, a substance or a radionuclide can be released. In this sense, the term is completely misleading and should be set in perspective.

Authors’ response: We thank the reviewer for these observations. In p.1 paragraph 1 the radioactivity is changed to radionuclide. Kindly see line 34.

What is the chemical form of radionuclides that can leave liposomes? Are they really free radionuclides, as stated on p. 2, paragraph 2? Or can labeled molecules also leave liposomes? Or will radiolysis occur due to nuclear reflection? Could you comment this?

Authors’ response: Liposomes are drug carrier vesicles; it does not affect or change the manner of the encapsulated agents. We also used cholesterol in liposome formation as it increase the encapsulation efficiency and stability over time leading to decrease the encapsulated drug leakage.

Please pay attention to the correct chemical nomenclature, according to IUPAC, especially writing the nomenclature of labeled molecules. Also, pay attention to the correct writing of indices for radionuclides - in many places this requires checking. The text thus has a very distracting effect and gives the impression of a hastily written manuscript.

Authors’ response: We agree and we thank the reviewer for these suggestions. The manuscript is edited as requested and corrected by MDPI English pre-edited services to ensure we did not miss any error.

e.g. 1,2-dipalmitoyl-sn-glycero-3-phosphocholine ... what does mean -sn-?

Authors’ response: Thank you for the suggestion. We agree it is appropriate to use it as 1,2-dipalmitoylphosphocholine (DPPC) (see line 95). Stereospecific numbering (sn) used in glycerophospholipids by numbering, stereospecifically, the carbon atoms of glycerol to designate the configuration of glycerol derivatives.

also should be 99mTc instead of 99mTc and in case of selectively labeled compounds ought to be [99mTc]. Also ionic form must be written in proper form.

Authors’ response: We agree, thank you for the observation. The manuscript was revised in regards.

Basic information about the activity of the prepared [99mTc]MIBI complex is missing in the paragraph on labeling. The radiochemical yield is not indicated either. It is completely unacceptable to state the unit of specific activity Mbq/Kg, instead of the correct form MBq/kg.

Authors’ response: Thank you for the corrections. Basic information about the activity of the prepared [99mTc]MIBI complex and radiochemical yield are added to the paragraph on labeling line (95). Also the units are corrected.

In paragraphs concerning the animal model: Could you specify a number of subject in animal cohort? Is it correct to perform data only on 5 animals? This is also necessary to comment it due to statistical evaluation of data.

Authors’ response: Thank you for the suggestion and we agree with the reviewer. This research was conducted by reducing the sample size to five rabbits that were imaged at different times and experimenting different types of radiopharmaceuticals through strategies within ethically acceptable bounds and without losing the statistical power. Kindly see line 170.

Could you please comment a physical meaning of the entrapment efficiency 99.9997% How did you carry on?

Authors’ response: The supernatant was collected for HPLC analysis to assess the drug loading capacity and 99.9997% of the tracer was entrapped within the liposomes.

On Figure 1 is chromatogram of liposomes without any hot label.

Authors’ response: Thank you for the suggestion. Hot label is added to Figure 1 as requested.

How did you carry on labeling or liposomal internalisation of [99mTc]MIBI? How you can be sure, that [99mTc]MIBI remains in liposome for certain time?

Authors’ response: We report the encapsulating efficiency of the tracer and radiotracer within liposomes (both of them got the same results), as we were testing the pH gradient technique effectiveness for the radiopharmaceuticals before applying it. We added cholesterol in liposomes formation to increase the entrapment of the loaded agent. As we ensure the effectiveness of the pH gradient technique, we precede this research to determine the best way possible to use the encapsulated tracer without any radioactive contaminations as rapid one-step kit for the routine preparations. Besides, in this study we mainly were testing the pharmacokinetics of two encapsulation techniques and other advantages of using liposomes loaded radiopharmaceutical agents. 

Evaluation of stability in vitro is missing.

Authors’ response: Thank you for the suggestion. We totally agree and this is a good idea to do it in future researches.

 In description of TLC the mobil phase and stationary phase are not listed as well as set up of detector and used collimator.

Authors’ response: Thank you for the suggestion. Kindly check labeling procedure paragraph, line 148.

The values in tables and graphs (Table 1, 2, 3 and Figure 3) are in counts. Should be more appropriate to show % of applied dose per mass of organ which is more comon for ex vivo. However, for SPECT would be interesting SUV and related parameters.

Authors’ response: Authors highly appreciate the reviewer’s suggestion. As it is not precise to get the injected dose percentage from planar images, thus we used the maximum count per organ to determine the biodistribution and clearance mechanism. SUV is a good point to rise as we use it for our next research in PET scan.

The size of used liposome is being described. Did you observe any changes of hydrodynamic ratio of Zeta potencial after internalisation of [99mTc]MIBI?

Authors’ response: We agree with the reviewer point of view and we confirm that the zeta potential of the liposomes increased after internalisation of MIBI as it is cationic tracer. Thus we did use less cationic particles in liposomes loaded MIBI.

The conclusion is insuficient. Unfortunately, the conclusion misses the key facts that were reached in the study. I recommend rewriting it.

Authors’ response: Thank you for the insightful suggestion. Conclusion has been revised and rewritten as requested.

Reviewer 3 Report

Drug delivery systems using different vehicles (nanoparticles, liposomes and others) are in the focus of attention for various fields of medicine, including advanced nuclear medicine techniques, such as gamma scintigraphy and single photon emission tomography (SPECT). For these methods technetium-99m (99mTc) has been by the most commonly used radionuclide to radiolabel liposomes, due to its wide availability from isotopic generators at low cost, favourable imaging properties,
and a half-life of about 6 h that allows imaging for up to 24 h. In the present study the authors suggested to modify a well-known cardiac SPECT agent,
[99mTc]MIBI via its encapsulation into positive charged liposomes, aiming improve uptake, clearance and biodistribution of the original “classical” [99mTc]MIBI formulation. For both 99mTc-liposomal agents, 99mTc-MIBI-free liposomes and 99mTc-liposomes loaded MIBI, radiolabelled within the study by different methods, the uptake was higher and the extraction was faster than [99mTc]MIBI control just after administration. At the same time at 1 h after administration fast clearance of the radioactivity was detected which was beneficial to reduce radiation dose for the patients. An important issue was the prevention of the radioactivity leakage from the encapsulated radiopharmaceutical that was achieved via pH-gradient radiolabeling technique.

In general, the novelty of the data and the results obtained is doubtless. However, I found reading the manuscript to be very difficult, especially in the Results and Discussion part. The abstract did not reflect all the findings of the study, it is too general and have to be reformulated and more tightly connected to the article idea and conclusions.   

In addition, there are some issues that are listed below in no specific order that should be addressed prior to publication.

  1. The study was performed in five male rabbits which were injected with all three radiotracers with a one week break between the studies. Is it allowable from ethical point of view, especially considering extremely high injected dose of 296 Mbq 99mTc-MIBI and 99mTc-liposomal agents diluted in 5 ml saline? (please note, it should be MBq, not Mbq)
  2. Why the injected dose was so high? In the previous article from the same group the injected dose was only about 44 MBq for positive-charged liposomes (Alkandari AM, Shafaa MW, Alsayed YM. Preclinical Assessment of 99mTc-Labeled Liposome Agents as an Effective Tracer in Nuclear Medicine. J Nucl Med Technol. 2020 Sep;48(3):269-273. doi: 10.2967/jnmt.119.239756.).

  1. The authors claimed that they reached high encapsulation efficiency during radiolabelling by both methods, however there was no illustrative materials to support. The only chromatogram (Fig. 1) demonstrated single peak which was not assigned to any compound. The Fig. 1 legend has to be reformulated. What does it mean “HPLC test using Kromasil® chromatogram”? Kromasil is a packing material.

4.      There is a question about determination of IC50 value. The authors wrote: “A toxicity test was also performed and the half-maximal-inhibitory-concentration (IC50) was found to be >100 µg/ml.” It would be desirable to provide the details and comments on the methodology employed by the authors and corresponding reference. What inhibitor was used in evaluation of the IC50? Also the  IC50 is usually expressed in micromoles or nanomoles, For example, the reported IC50 value for  [99mTc]MIBI inhibited by a metabolic inhibitor, iodoacetate, was 5 x 10-6M (Piwnica-Worms D., Kronauge J.F., Delmon L., Holman B.L., Marsh J.D., Jones A.G. Effect of metabolic inhibition on technetium-99m-MIBI kinetics in cultured chick myocardial cells. J Nucl Med. 1990;31(4):464–72).

            How to explain such a big discrepancy in the values?

  1. The text of the manuscript has to be checked by native English-speaking person or by editors, both for grammar and numerous type-mistakes (Mereck Millipore, Mbq, “The flow rate was 1 ml/min and used a 260nm UV detector”, etc.).

As such, I would recommend this paper for publication after considering above questions and comments.

Author Response

Response to Reviewers’ Comments

Journal: Biomedicines

Manuscript number: biomedicines-2016488

Manuscript title: Enhanced Efficacy of Radiopharmaceuticals When Using Technetium-99m-Labeled Liposomal Agents: Synthesis and Pharmacokinetic Properties.

Dear Respected Editors/ Reviewers

We appreciate you for your precious time in reviewing our manuscript and providing valuable suggestions. Also we are grateful for the insightful comments to improve our manuscript and we have been incorporate changes to reflect the suggestions requested. Here is point-by-point response to the reviewer concerns. Kindly see the attachment for the edited manuscript. 

In general, the novelty of the data and the results obtained is doubtless. However, I found reading the manuscript to be very difficult, especially in the Results and Discussion part. The abstract did not reflect all the findings of the study, it is too general and have to be reformulated and more tightly connected to the article idea and conclusions.   

Authors’ response: We appreciate the reviewer’s consideration and we hope that the revised manuscript is satisfactory. Abstract and conclusion are edited as requested.

In addition, there are some issues that are listed below in no specific order that should be addressed prior to publication.

  1. The study was performed in five male rabbits which were injected with all three radiotracers with a one week break between the studies. Is it allowable from ethical point of view, especially considering extremely high injected dose of 296 Mbq99mTc-MIBI and 99mTc-liposomal agents diluted in 5 ml saline? (please note, it should be MBq, not Mbq)

Authors’ response: We thank the reviewer for the kind reminder. We totally agree that the use of animals in research should be ethically and morally justified in order to prevent undue suffering. Thus, before experiments are conducted, animal ethics committee assigned one week break between the studies. Also in a medical point of view one week was suitable for the injected radioactivity to be cleared out.

  1. Why the injected dose was so high? In the previous article from the same group the injected dose was only about 44 MBq for positive-charged liposomes (Alkandari AM, Shafaa MW, Alsayed YM. Preclinical Assessment of 99mTc-Labeled Liposome Agents as an Effective Tracer in Nuclear Medicine. J Nucl Med Technol. 2020 Sep;48(3):269-273. doi: 10.2967/jnmt.119.239756.).

Authors’ response: We agree. In the previous article we were injecting the rabbits subcutaneously in the dorsum of each hind foot over the region of the metatarsals at the midline, however in this research rabbits were injected intravenously. Also the age and weight of the rabbits of the previous study were half of the age and weight of the rabbits in this research as the marginal ear vein is hardly visualized in younger rabbits. Besides the delayed images of the previous study was only at one hour post injection compared to this study which imaged rabbits up to 24 hrs post injection.

  1. The authors claimed that they reached high encapsulation efficiency during radiolabelling by both methods, however there was no illustrative materials to support. The only chromatogram (Fig. 1) demonstrated single peak which was not assigned to any compound. The Fig. 1 legend has to be reformulated. What does it mean “HPLC test using Kromasil® chromatogram”? Kromasil is a packing material.

Authors’ response: We thank the reviewer for the suggestion. Both agents were tested and got the nearly the same peaks. However, the chromatogram in the present study identifies the encapsulation efficiency of MIBI encapsulated within liposomes as the radiolabeling was acquired after the encapsulation process. The main purpose is to get ready-to-use kit that can be easily applicable in daily routine without causing any radioactive contamination of the instruments, thus we found that pH gradient technique is the best method followed by lyophilization. Fig.1 legend has been reformulated as requested, kindly see line 212.

  1. There is a question about determination of IC50 value. The authors wrote: “A toxicity test was also performed and the half-maximal-inhibitory-concentration (IC50) was found to be >100 µg/ml.” It would be desirable to provide the details and comments on the methodology employed by the authors and corresponding reference. What inhibitor was used in evaluation of the IC50? Also the  IC50 is usually expressed in micromoles or nanomoles, For example, the reported IC50 value for  [99mTc]MIBI inhibited by a metabolic inhibitor, iodoacetate, was 5 x 10-6M (Piwnica-Worms D., Kronauge J.F., Delmon L., Holman B.L., Marsh J.D., Jones A.G. Effect of metabolic inhibition on technetium-99m-MIBI kinetics in cultured chick myocardial cells.J Nucl Med. 1990;31(4):464–72).

            How to explain such a big discrepancy in the values?

Authors’ response: We thank the reviewer for pointing this out. We totally agree about the concept of converting the concentration to nanomoles to study the biological effect of specific dose as reported in (Piwnica-Worms D., Kronauge J.F., Delmon L., Holman B.L., Marsh J.D., Jones A.G. Effect of metabolic inhibition on technetium-99m-MIBI kinetics in cultured chick myocardial cells. J Nucl Med. 1990;31(4):464–72). However, we encapsulate the sestamibi in liposomes, different molecules were added, and thus the molecular weight is different comparing to sestamibi without liposomes. Besides, in this research we studied the mass dose to format applicable encapsulated tracer. Kindly see the example (Rossano, S.; Naganawa, M.; Finnema, S.; De Bruyn, S.; Otoul, C.; Stockis, A.; Nicolas, J.-M.; Martin, P.; Maguire, R.; Mercier, J.; Carson, R. Estimation of SV2A Occupancy and IC50 of Antiepileptic Drugs Levetiracetam and Brivaracetam Using 11C-UCB-J PET. J Nucl Med 201859 (supplement 1), 1720.)

  1. The text of the manuscript has to be checked by native English-speaking person or by editors, both for grammar and numerous type-mistakes (Mereck Millipore, Mbq, “The flow rate was 1 ml/min and used a 260nm UV detector”, etc.).

Authors’ response: We agree and we thank the reviewer for this observation. The manuscript it edited as requested then corrected by MDPI English pre-edited services to ensure we did not miss any error.

Reviewer 4 Report

This study aimed to formulate suitable liposomes using different techniques and compounds for enhanced stability, uptake, and clearance based on Technetium ( Tc) sestamibi, a lipophilic cation that, when injected intravenously into a patient, distributes in the myocardium proportionally to the myocardial blood flow. Authors tested three samples, 99mTc-MIBI without liposomes (control), 99mTc-MIBI added to a lower pH liposome (99mTc-MIBI-free liposomes gradient technique) and MIBI encapsulated during the formulation of the liposome then labelled with 99mTc (99mTc-liposomes loaded MIBI).

In their summary, they concluded that liposomes show particular promise in intracellular delivery systems and drug delivery to specific locations. Additionally, they have shown that encapsulating MIBI with liposomes enhanced the biodistribution, uptake, and clearance of radiopharmaceuticals, which may reduce their toxicity but at the same time preparation must be carried out at the imaging time to avoid leakage from the liposomes and ensure their stability.

The manuscript required some minor revisions:

Introduction

Radionuclide imaging is obtained not only by gamma camera but also by PET scanner. Therefore, there should be a general sentence in the introduction, or the author should narrow imaging down to technetium imaging.

Analytical method:

The abbreviation for Acetonitrile should be MeCN (methyl cyanide), not CAN.

What is the origin of (H9C2) rat heart/myocardium cells?

The first occurring abbreviation has to be preceded by a full name (SRB assay).

Authors should pay more attention to the spelling and abbreviations used: SI units MBq (not Mbq), kg (not Kg), mL (not ml), 99mTc, free pertechnetate 99mTcO4- (during RCP, there is an ion of pertechnetate).

Please state exactly which company the gamma camera was from.

According to journal requirements, authors must include housing, husbandry and pain management details in their manuscript.

Results

What method was used for statistical analysis?

What are the units in Tables 1, 2 and 3 for biodistribution? Is it counts±SEM?

Since images were taken 24 h after injection, authors should present the in vitro stability data of 99mTc-MIBI, 99mTc-MIBI-free liposomes and 99mTc-loaded liposomes MIBI.

There are no RCP results of labelled compounds.

In figure 5 a) and b), the complexes must be described by the same colours, otherwise, the reader is misled (not intentionally).

Abbreviations:

My recommendation: it is easier to read/ search when the abbreviations are placed in alphabetic order.

In conclusion, reading the article, one feels the lack of images from animal SPECT camera, which would have made the publication more attractive. The main aim of the studies performed was reached, therefore, I do recommend the manuscript for publication after minor revision.

Author Response

Response to Reviewers’ Comments

Journal: Biomedicines

Manuscript number: biomedicines-2016488

Manuscript title: Enhanced Efficacy of Radiopharmaceuticals When Using Technetium-99m-Labeled Liposomal Agents: Synthesis and Pharmacokinetic Properties.

Dear Respected Editors/ Reviewers

The authors are highly appreciate the comprehensive feedback throughout the review process. We also would like to thank the reviewers for all useful and helpful comments that improved the manuscript.

Comments and Suggestions for Authors

The manuscript required some minor revisions:

Introduction

Radionuclide imaging is obtained not only by gamma camera but also by PET scanner. Therefore, there should be a general sentence in the introduction, or the author should narrow imaging down to technetium imaging.

Authors’ response: We agree and thank you for the observation. PET scanner is added as requested. Kindly see line 34.

Analytical method:

The abbreviation for Acetonitrile should be MeCN (methyl cyanide), not CAN.

Authors’ response: Thank you for the suggestion. Changed as requested, kindly see line 113.

What is the origin of (H9C2) rat heart/myocardium cells?

Authors’ response: Thank you for your observations and corrections. The origin of (H9C2) rat heart/myocardium cells is from American Type Culture Collection ,ATCC, Manassas, USA. It is also has been added in Cytotoxic activities section, line 118.

The first occurring abbreviation has to be preceded by a full name (SRB assay).

Authors’ response: Thank you for the suggestion. Edited as requested. Kindly see line 122.

Authors should pay more attention to the spelling and abbreviations used: SI units MBq (not Mbq), kg (not Kg), mL (not ml), 99mTc, free pertechnetate 99mTcO4(during RCP, there is an ion of pertechnetate).

Authors’ response: Thank you for the correction. Kindly see lines 143,156, and 157.Also the manuscript was revised in regards as requested and double checked by MDPI English pre-edited services to avoid any error.

Please state exactly which company the gamma camera was from.

Authors’ response: Thank you for the suggestion. Gamma camera was from (Symbia, Siemens health care, Illinois, USA). This information is added to the first mention in the Dosimetry and imaging protocol section, line 159.

According to journal requirements, authors must include housing, husbandry and pain management details in their manuscript.

Authors’ response: We agree. The animal model section is edited as requested. Kindly see line 167.

Results

What method was used for statistical analysis?

Authors’ response: Descriptive statistics and predictive statistical regression method.

What are the units in Tables 1, 2 and 3 for biodistribution? Is it counts±SEM?

Authors’ response: That is correct, the units used for tables 1,2 , and 3 were counts±SEM.

Since images were taken 24 h after injection, authors should present the in vitro stability data of 99mTc-MIBI, 99mTc-MIBI-free liposomes and 99mTc-loaded liposomes MIBI.

Authors’ response: Thank you for the insightful suggestion. We agree and we will ensure to add it to future researches.

There are no RCP results of labeled compounds.

Authors’ response: Thank you for the suggestion. Edited in Labeling procedure section. Kindly see line 146.

In figure 5 a) and b), the complexes must be described by the same colours, otherwise, the reader is misled (not intentionally).

Authors’ response: We agree. Changes applied to figure 5 and the manuscript was revised in regards. Kindly see line 344.

Abbreviations:

My recommendation: it is easier to read/ search when the abbreviations are placed in alphabetic order.

Authors’ response: We agree and appreciate your observation. The abbreviations are alphabetically ordered and adjusted according to this comment. Kindly see line 424.

In conclusion, reading the article, one feels the lack of images from animal SPECT camera, which would have made the publication more attractive. The main aim of the studies performed was reached, therefore, I do recommend the manuscript for publication after minor revision.

Authors’ response: The authors would like to thank the reviewer for the positive feedback on our manuscript. All comments have been taken into account and the paper has been revised accordingly.

Round 2

Reviewer 2 Report

The manuscript has been revised and edited to a sufficient extent. Data that was not included in the previous version was added. The editing was carried out to a sufficient extent, therefore I recommend the manuscript for acceptance.

Reviewer 3 Report

Dear authors, thank you for a very clear answering my questions and revisions made in the text of the manuscript. Still I can see some typos. Plese make double check. For example, in conclusion "enchantment" should be replaced with "enchancement". In the introduction please check the sentence "The radionuclide is cleared from the body by the radioactive decay and the biological clearance, the effective half-life (2)." - it looks like something is missing in this sentence.